# Cheminformatics and Machine Learning Approaches to Assess Aquatic Toxicity Profiles of Fullerene Derivatives

**DOI:** 10.3390/ijms241814160

**Published:** 2023-09-15

**Authors:** Natalja Fjodorova, Marjana Novič, Katja Venko, Bakhtiyor Rasulev, Melek Türker Saçan, Gulcin Tugcu, Safiye Sağ Erdem, Alla P. Toropova, Andrey A. Toropov

**Affiliations:** 1Laboratory for Chemoinformatics, Theory Department, National Institute of Chemistry, Hajdrihova 19, 1001 Ljubljana, Slovenia; marjana.novic@ki.si (M.N.); katja.venko@ki.si (K.V.); 2Department of Coatings and Polymeric Materials, North Dakota State University, NDSU Dept 2510, P.O. Box 6050, Fargo, ND 58108, USA; bakhtiyor.rasulev@ndsu.edu; 3Ecotoxicology and Chemometrics Lab, Institute of Environmental Sciences, Bogazici University, Hisar Campus, 34342 Istanbul, Turkey; msacan@boun.edu.tr; 4Department of Toxicology, Faculty of Pharmacy, Yeditepe University, Atasehir, 34755 Istanbul, Turkey; gulcin.tugcu@yeditepe.edu.tr; 5Department of Chemistry, Marmara University, 34722 Istanbul, Turkey; erdem@marmara.edu.tr; 6Laboratory of Environmental Chemistry and Toxicology, Istituto di Ricerche Farmacologiche Mario Negri, Via Mario Negri 2, 20156 Milano, Italy; alla.toropova@marionegri.it (A.P.T.); andrey.toropov@marionegri.it (A.A.T.)

**Keywords:** fullerene-based nanomaterials, fullerene derivatives, artificial neural network, aquatic toxicity, protein–ligand binding activity, binding affinity, CORAL software, ToxAlerts

## Abstract

Fullerene derivatives (FDs) are widely used in nanomaterials production, the pharmaceutical industry and biomedicine. In the present study, we focused on the potential toxic effects of FDs on the aquatic environment. First, we analyzed the binding affinity of 169 FDs to 10 human proteins (1D6U, 1E3K, 1GOS, 1GS4, 1H82, 1OG5, 1UOM, 2F9Q, 2J0D, 3ERT) obtained from the Protein Data Bank (PDB) and showing high similarity to proteins from aquatic species. Then, the binding activity of 169 FDs to the enzyme acetylcholinesterase (AChE)—as a known target of toxins in fathead minnows and *Daphnia magna*, causing the inhibition of AChE—was analyzed. Finally, the structural aquatic toxicity alerts obtained from ToxAlert were used to confirm the possible mechanism of action. Machine learning and cheminformatics tools were used to analyze the data. Counter-propagation artificial neural network (CPANN) models were used to determine key binding properties of FDs to proteins associated with aquatic toxicity. Predicting the binding affinity of unknown FDs using quantitative structure–activity relationship (QSAR) models eliminates the need for complex and time-consuming calculations. The results of the study show which structural features of FDs have the greatest impact on aquatic organisms and help prioritize FDs and make manufacturing decisions.

## 1. Introduction

Carbon nanomaterials (CNMs) are organic compounds with a size of 1–100 nm. Fullerenes, also called buckyballs, are spherical molecules composed of carbon atoms [1]. The unique physical and electrochemical properties of CNMs and products based on them have ensured their wide application. Fullerene derivatives (FDs) are among the most popular CNMs that have found wide application [2]. They are used in materials science [3,4] and the cosmetics and pharmaceutical industries [5]. Although the original fullerene C60 is practically insoluble in polar solvents, the addition of functional groups to C60 opens up numerous applications and increases its bioactivity [6,7,8]. Therefore, FDs have a wide range of applications in medicinal chemistry [9]. They have therapeutic characteristics, being antioxidant [10], cytoprotective against oxidative stress [11], anti-influenza [12], antimicrobial [13], and antiviral [14]. They are also used as carriers in drug delivery systems [15,16].

The increasing occurrence of FDs, due to their benefits, can potentially disrupt aquatic environments. In addition to the direct hazards of FDs [17], other ecotoxicological aspects, such as endocrine-disrupting effects, interactions with other chemicals, and (bio)accumulation make them pollutants of environmental concern [8]. An assessment of their toxicity to aquatic organisms contributes to the risk assessment of FDs [2,18,19]. The acute toxicity and sub-lethal effects of FDs on various aquatic organisms were evaluated in a study by Oberdörster et al. [20]. Fish are one of the model organisms for aquatic ecosystems. Studies on the potential toxicity of FDs to fish at different levels (molecular, cellular, tissue, organ, etc.) can be found in the literature. FDs have been shown to damage gonadal tissues in freshwater fish [21]. Other effects of FDs on fish organs include those affecting the brain [22], gills, and liver [23]. The formation of reactive oxygen species (ROS) by C60, which causes oxidative stress in fish, has been reported in several studies in the literature [24,25]. In a study using *Danio rerio* as a test organism, fullerene exposure resulted in delayed embryo and larval development, reduced survival and hatching rates, and pericardial edema in zebrafish [24]. The same study suggests that the developmental damage was caused by a process triggered by free radicals or another type of oxidative stress [24]. In contrast, another study reported by Henry et al. [26] states that the formation of ROS is minimal. Nevertheless, the increasing use of FDs in many different fields, their potential spread in the environment, and the evidence that FDs have some adverse effects on fish indicate that further toxicological studies in water are needed [27].

After their use in various applications, FDs are released into the aquatic environment through wastewater treatment plants and effluents. It has been reported that wastewater treatment plants may not be able to effectively remove these chemicals from wastewater [2,19]. The median effective concentration (EC50) of chemicals that are harmful is typically reported in mg/L and can differ significantly depending on various factors such as the type of aquatic organism, the duration of exposure, and the utilization of different chemical preparation methods. On the other hand, the measured concentrations of FDs in aquatic environments, including wastewater treatment plants, have been calculated to have a mean concentration of 567 ng/L [2]. Therefore, toxicity studies based only on EC_50_ values do not reflect all aquatic toxic effects of chemicals at low concentrations. When an organism is exposed to a low concentration of a chemical, toxic effects occur when the chemical binds to specific proteins in the organism. Hence, it is important to investigate their propensity to bind to proteins of aquatic species using computational methods to understand the effects of low-level FDs on non-target aquatic organisms. Unfortunately, there is no information in the literature on the PDB structures of proteins involved in aquatic toxicity. However, a study by McRobb et al. [28] that compared orthologues in aquatic species with protein structures and sequences that are critical for human toxicity may be useful in identifying toxicity in aquatic species. In their study, they examined the overall sequence similarity of human proteins to various aquatic organisms and found that human proteins averaged 69% similarity to *Danio rerio* (zebrafish), 63% similarity to *Pimephales promelas* (flat-headed minnow), 70% similarity to *Takifugu rubripes* (Japanese pufferfish), 71% similarity to *Xenopus laevis* (African clawed frog), and 72% similarity to *Xenopus tropicalis* (west clawed toad). They also reported an overall sequence similarity of 88% between the human androgen receptor and *X. tropicalis.*

In drug safety assessment and environmental toxicology, some organisms such as rodents and fish [29,30] are mainly used as model organisms for the toxicological characterization of chemicals, and an extrapolation approach has been used [31]. Extrapolation across endpoints allows for animal-free assessment and a better use of toxicity studies [32].

Computational methods have been developed over many years and play a critical role in estimating the aquatic toxicity of both new and existing compounds [33,34,35,36,37,38,39]. Quantitative structure–activity relationships (QSARs) are an effective method for predicting the aquatic toxicity of chemicals [40,41,42,43,44,45]. Fullerene-specific QSAR applications have been used for various endpoints and properties, particularly for binding activities [46], solubility [47,48], and cytotoxic effects [49]. Integrated approaches in toxicology include combining in silico and in chemico methods to reduce animal testing [50,51,52]. In this context, the modeling of the binding activities of compounds to toxicity-related enzymes is of great importance for aquatic toxicity and has a place in regulatory guidelines. Machine learning approaches, such as counter-propagation artificial neural networks (CPANNs), are used as effective methods for aquatic toxicity profiling [53,54].

QSAR models are based on the study of the relationships between the structure and properties of compounds. In this study, the property of compounds was expressed as the binding affinity between FDs and human proteins, which have high similarity with aquatic species. In the previous receptor- and ligand-based study of fullerene analogs, using a computational approach including quantum-chemical, QSAR and molecular docking simulations, performed by Ahmed et al. [55], a high correlation between calculated and experimental binding affinities (binding energy (BE)) was demonstrated for 20 FDs. Therefore, in our study, the hypothesis was based on the assumption that the binding affinity of fullerenes to proteins is associated with toxicity (aquatic toxicity). The aquatic toxicity of fullerene derivatives (FDs) was evaluated using the binding affinity of FDs to ten proteins similar to aquatic species as well as to acetylcholinesterase enzyme (AChE), with a known toxic mode of action. We complemented the QSAR study by analyzing structural alerts (SAs), which allowed us to obtain additional information about fragments associated with the aquatic toxicity of the studied molecules. A web server, ToxAlerts [56], was used to collect SAs for aquatic toxicity.

The application and study of structural alerts for aquatic toxicity have been reported in recent papers. For example, the article by Zhang et al. [57] reported on the structural basis of acute aquatic toxicity. Khan et al. [58] proposed fragment-based descriptors for QSAR modeling of aquatic toxicity. Tinkov et al. [59] published an article discussing how the acute aquatic toxicity of organic compounds can be influenced by structural patterns.

SAs also have the potential to be used in conjunction with other experimental methods to evaluate the endocrine disruption effects of chemicals [60]. Prioritization for further evaluation can be supported in a tiered approach with evaluation of all in silico methods.

In our previous work [46], we analyzed the structural features that have an impact on the binding affinity of 169 FDs to 1117 human proteins. The next step in the study of the properties of FDs was devoted to the influence of FDs on important targets related to diabetic disease [61]. The aim of this study is to evaluate the aquatic toxicity of FDs. We selected the structures of human proteins that share similarities with aquatic species-specific proteins related to the endocrine system (androgen and estrogen receptor (AR and ER1)), neurotransmitter oxidation (monoamine oxidase-A (MAO-A) and monoamine oxidase-B (MAO-B)), metabolism-related enzymes (CYP3A4, CYP2C9, and CYP2D6), and others. The 10 human proteins (1D6U, 1E3K, 1GOS, 1GS4, 1H82, 1OG5, 1UOM, 2F9Q, 2J0D, and 3ERT) associated with aquatic toxicity were taken from the Protein Data Bank (PDB). Protein binding affinity data were calculated using several bioinformatics methods. First, we calculated the binding affinity for 169 FDs associated with the above 10 proteins. Then, the determined binding affinity for these 10 proteins was compared with the average binding affinity for all 1117 proteins (referred to as average binding scores (Average BScores)). The novelty of our study is that we investigated aquatic toxicity using the binding affinity of FDs for human proteins similar to those of aquatic species.

In addition, we focused on possible mechanisms of action between FDs and proteins. As reported in the article by Tinkov et al. [59], the reactions of nucleophilic substitution, inhibition of oxidative phosphorylation, interaction with acetylcholinesterase and voltage-gated ion channels are among the most important mechanisms of the toxic effect of compounds on fathead minnow, *Daphnia magna* and *Tetrahymena pyriformis*. The mechanism of toxic action based on the inhibition of acetylcholinesterase (AChE) by toxic compounds was described in the work of Ćolović et al. [62]. Mechanistic insights from integrated ligand-/structure-based assessments on *Tc*AChE were presented in the work of Minovski et al. [48]. The authors investigated the role of pharmaceutical compounds in fish toxicity. They suggested that the binding of pharmaceutical compounds to *Tc*AChE may have the potential to trigger molecular initiating events for adverse outcome pathways (AOPs).

The aim of our study was to investigate the effect of FDs on AChE and its possible contribution to aquatic toxicity. We used *Tc*AChE (acetylcholinesterase from *Torpedo californica*) as a model protein and calculated the binding affinities of 169 FDs to this enzyme using PDB code 1VOT. The results of our analysis showed that FDs can inhibit the activity of *Tc*AChE, consistent with their potential role in aquatic toxicity. To further support our results, we also used the aquatic toxicity SAs collected in the ToxAlerts server [56]. Our results provide new insights into the mechanism of aquatic toxicity and suggest the potential role of FDs in this process.

The QSAR models were built based on regression and neural network algorithms according to the five OECD principles for QSAR models, which were accepted for regulatory purposes in 2007. In this study, machine learning, including neural network algorithms, was applied to predict the binding affinity of FDs. Cheminformatics tools enabled the analysis of chemical space to determine the features responsible for the toxic effects of the compounds considered. These models can be used to rank FDs in regulatory toxicology, prioritize compounds, and reduce animal testing.

## 2. Results and Discussions

### 2.1. The Modeling Plan Carried Out in the Study

This study used regression models and models based on the CPANN algorithm. This allowed for a double-check of the models and the results obtained. Information about the models created in the study is shown in Table 1.

The use of different prediction algorithms (regression and CPANN) represents a double-check of the models on the one hand, and on the other hand, it confirms the reliability of the prediction methods when we have obtained identical results.

The models, which included all 28 descriptors, aimed to find (1) highly correlated variables (between descriptors and/or property = binding affinity) and (2) the most significant features affecting the binding affinity. The top map of this model (as a visualization tool) was supplemented with information related to *Tc*AChE inhibitory activity and SAs related to aquatic toxicity for the most and least active FDs.

The best optimal models for predicting binding affinity based on two descriptors (QPpolrz + TD) and/or based on optimal descriptors (DCW) aim to predict binding affinity and avoid lengthy calculations for new FDs with unknown affinity.

### 2.2. ALL_CPANN Model Based on Twenty-Eight Descriptors

CPANN models based on twenty-eight descriptors (twenty-five drug-like descriptors plus QPpolrz, TD and DCW descriptors) were built. We trained the optimal model with an architecture of 20 × 20 neurons for 600 epochs. The statistical performance of the CPANN model, referred to as the ALL_CPANN model, is shown in Appendix A. The model was characterized by a coefficient of determination R^2^ = 0.995–0.965 and a root-mean-square error RMSE = 0.068–0.188 with respect to the outcome variables considered.

In this study, a self-organizing Kohonen map model was used, including a top map showing the distribution of FDs in a 2D map, weight maps for each descriptor, and the distribution of binding affinity values for each outcome variable. The CPANN model was used as a look-up table to compare clusters of FDs with similar structures and values for specific descriptors and binding affinities by examining the position of similar objects on the Kohonen maps.

### 2.3. Investigation of the Relationship between the Studied Binding Activities in Regression and ANN Models

The relationships between all the studied binding affinities were investigated in the ALL_CPANN model and ALL_regression model based on twenty-eight descriptors.

First, we examined the correlations and similarities between the output variables considered: Average Binding Score (Average BScore), binding affinity, and binding scores for ten proteins relevant to ten aquatic species.

The output maps for all these output variables are shown in Figure 1.

The dark-red color indicates the highest binding affinity and corresponds to the location of the most active FDs on the top map. The dark-blue color refers to the lowest values and corresponds to the location of the least active FDs on the top map. The similarity of the color-coded weight maps in Figure 1 confirms a relationship between the binding affinities of all selected variables (from (a) to (l)). Thus, the weight map of Average BScore (a) has a similar distribution of the highest and lowest values to the weight maps of binding affinity (b) and binding scores for ten proteins associated with aquatic species (from (c) to (l)). We can assume that all these values are highly correlated.

In the regression analysis, the statistical program Minitab was used to calculate Pearson correlation coefficients between all output variables in relation to binding affinity. The summary of results in Appendix A shows the range of correlation coefficients describing the relationship between the binding score of ten proteins associated with aquatic species, the average binding score, and the binding affinity. The table confirms that there is a high correlation between the analyzed variables. For example, the correlation coefficients between the binding scores of ten proteins associated with aquatic toxicity and the Average Binding Score ranged from 0.966 to 0.656, while the correlation coefficients between the binding scores of ten proteins associated with aquatic toxicity and the binding affinity ranged from 0.968 to 0.669. In addition, the correlation between the binding scores of ten proteins was found to range from 0.937 to 0.570.

Both methods (regression and CPANN) yielded identical results in terms of high correlation.

In the next step of our study, the Average Binding Scores were used as a reference to compare with the binding scores of 10 proteins related to aquatic toxicity and with the binding scores for the protein 1VOT, related to *Tc*AChE.

Figure 2 shows a summary plot comparing the binding scores of ten proteins associated with aquatic toxicity with the Average Binding Scores. The binding scores for each of the ten proteins studied, compared separately with the Average Binding Score, are shown in Appendix A. The numbering of the FDs along the X axis is done in such a way that it reflects the arrangement of the FDs in ascending order of Average Binding Score values, with the first number corresponding to FD168 with the lowest binding scores and so on.

In Appendix A, it can be seen that the values of the binding scores for 1UOM and 1GS4 (proteins associated with the endocrine system) are smaller than the values of Average Binding Scores, whereas the binding scores values for the remaining eight proteins are larger than the Average Binding Scores.

In addition, we constructed a plot Appendix A to compare binding scores for protein 1VOT, associated with *Tc*AChE, with the Average Binding Scores. This plot shows that most of the values of the binding scores for 1VOT (*Tc*AChE) are slightly higher than the Average Binding Scores.

### 2.4. Investigating the Relationship between Considered Binding Affinities and Descriptors in Regression and ANN Models

The next objective of the study was to determine the correlation between the descriptors used and the binding affinities. This part of the study was based on the ALL_CPANN model and ALL_regression model based on twenty-eight descriptors. Because the Average BScores and binding affinities are correlated with the binding scores for ten considered proteins, we selected the Average BScores and binding affinities as a unifying reference feature. Thus, we chose the most similar weight maps for descriptors and output maps for Average BScores and binding affinity (Figure 3).

The output maps of Average BScore and binding affinities show a similar distribution of highest and lowest values as the weight maps for the considered descriptors: DCW (a), QPpolrz (b), TD (c), Total Surface Area (d), Molecular Weight (e), Rotatable Bonds (f), and Non-H-Atoms (g). This could be an indication that these descriptors reflect the most important features affecting binding affinity per se.

The results of the regression analysis are presented in Appendix A. This summary matrix with the ranges of correlation coefficients describes the relationships between the binding scores of ten proteins associated with aquatic species and descriptors. The correlation between the binding activities of ten proteins associated with aquatic toxicity and the descriptors (Non-H-Atoms, Rotatable Bonds, Molecular Weight, Total Surface Area, TD, QPpolrz, and DCW) ranged from 0.937 to 0.443.

Both methods (regression and CPANN) showed similar results.

In the next step, we built regression models with equations for all the output variables of the responses to identify the most important descriptors that influence binding affinity. The results are presented in Appendix A. Following this table, we discuss the main characteristics that most strongly influence the binding affinity relying on the results of regression analysis.

The statistical performance of the regression models (ALL_regression model) was characterized by the coefficient of determination *R*^2^ and is presented in Appendix A. In the case of the average BScore, *R*^2^ = 0.968. For binding affinity, we obtained *R*^2^ = 0.952. For the binding scores of ten proteins, *R*^2^ ranged from 0.646 (in the case of 1D6U) to 0.934 (for 2JOD).

Topological diameter (TD) was found to have the greatest influence on binding affinity. Except for 1H82, 2J0D and 3ERT, the size of FDs plays a crucial role in all responses. After TD, QPpolrz and DCW follow in terms of significance. The Molecular Weight and Non-H Atoms also appear to be important in most cases, except for 1GS4, 1H82, and some others.

### 2.5. Application of CPANN Tools to Analyze the Most and Least Active FDs Based on Structural Alerts

We examined the CPANN Kohonen maps to analyze the most and the least active FDs, (see Figure 4a and Figure 4b, respectively). These figures show the arrangement of FDs in the 20 × 20 top map of the CPANN model in combination with the distribution of values of Average BScore in the output layer. They were used to investigate the correlation between areas in a 2D map related to the arrangement of FDs with specific functional groups and areas relevant to a specific level of binding affinity. Based on the high correlation between the Average BScore, binding affinity, and binding scores of ten proteins associated with aquatic toxicity, we can conclude that the least active and most active FDs here are approximately the same for all responses. Therefore, we selected the top map showing the distribution of Average Binding Scores in subsequent studies.

The most active FDs are located in the red and orange-yellow regions, respectively, in Figure 4a. As mentioned earlier, one of the main mechanisms of toxic action is based on AChE (its inhibition by toxicants is described in the work [62]). In Figure 4a, we have marked the binding scores (BSc.) for 1VOT (*Tc*AChE), as they express the acetylcholinesterase (AChE) inhibition properties of FDs.

FDs with similar structural alerts for aquatic toxicity (marked as TA) were grouped into separate groups (1–9). These TAs were identified here. The detailed structure of the FDs from these groups and their meaning are listed in Appendix A. These groups from 1 to 9 were also labeled in the Kohonen 20 × 20 top map. The structures of the TAs are shown in Figure 4a. For each group, a list of FDs with an indication of values of Binding Scores (BSc.) for *Tc*AChE is shown.

Group 1 in the Kohonen top map (Figure 4a), located in the upper left corner, corresponds to the highest values of binding scores. This area is colored red-orange-yellow. It belongs to FDs 112, 113, and 154–157 with BScores = 7298–6846 and contains *TA662* (aliphatic secondary and tertiary amines) and *TA665* (primary alkyl amines). The structure of these FDs is shown in Appendix A. Let us consider the structures of this group of FDs. All these FDs contain pyrrolidine (5-membered ring) attached to the C60 core, and all contain aromatic nitrogen. We hypothesize that these groups may also support the binding affinity and thus toxicity of the FDs.

Group 2 is located to the right of Group 1 in the Kohonen top map in Figure 4a. It contains *TA642* (ammonium NH_3_^+^ groups) (Appendix A). This group includes the most active FDs (red color) with BScores ranging from 7514 to 6970. FD161–163 from this group contains between four and eight NH_3_^+^ groups.

Group 3 is located in the lower right part of the Kohonen map and contains nitro groups. We found the following structural aquatic toxicity alerts from ToxAlerts: *TA628—*β-unsaturated nitro compounds, *TA11521—*4-nitrobenzene, and *TA667—*activated alkenes and alkynes. In Appendix A, we show the most active FDs in Appendix A. Here, FDs containing from 6 to 14 –NO_2_ with BScores = 6054–6934. Appendix A shows FDs containing from one to three –NO_2_ with BScores = 6054–6426. The presence of F-groups (FD142) or bridged bicycle rings (FD140) leads to increasing values of binding scores (Appendix A).

The lower part of the Kohonen map lists the following groups: Group 4: FDs with phosphonate groups, including *TA*617—phosphoric acid ester, with BScores = 6856–7442 (see Appendix A); Group 5: FDs with halide groups (F) *TA634*, halogenated benzylic group *TA638* and unsaturated nitro groups *TA628* for FD 96 and 142 with BScores = 6538–6934 (see Appendix A); Group 6: FDs with nitrile groups *TA626*—α,β-unsaturated nitriles with BScores = 6068–6308 (see Appendix A).

Group 7 (FD35, 36 and 142) contains *TA667*—activated alkenes and alkynes and *TA631*—α, β-unsaturated carboxylic acid with BScores = 7752–6886. See Appendix A.

Group 8 (FD2 and FD55) belongs to the active FD located on the upper right side of the Kohonen map in Figure 4a with BScores = 6730–6646. This group contains alkylamines *TA665*. See Appendix A.

Group 9 (FD4, 6 and FD55) belongs to the most active FDs in the upper center of the Kohonen map in Figure 4a (orange color) with BScores = 7090–6982. They contain *TA1181*—carboxylic acid secondary amides, *TA1176*—carboxylic acids and *TA659*—aliphatic alcohols. See Appendix A.

We found FD17 to have high binding affinity (BScores = 7328), but we found no structural alert for aquatic toxicity for this FD. In Figure 4a, FD17 is located in the upper part of the top map in the middle, near Group 9. The structure of this FD is shown in Appendix A. FD17 has a complicated structure, and no SA for aquatic toxicity was found for this compound. However, due to its high value of binding scores, we included this FD in the list of potentially toxic compounds.

Most FDs with high values of BScores for *Tc*AChE contain structural aquatic toxicity alerts (TAs). However, affinity often depends on a combination of different functional groups. In any case, we can offer to use BScores for the preliminary assessment of the toxicity of FDs.

It has been observed that FDs with the same SA for aquatic toxicity may have different BScore affinity. In such cases, it is useful to use BScores that are close to the structural alerts to identify the more toxic compounds.

The least active FDs are located in the blue and dark-blue areas of the 2D map, associated with the lowest values of Average BScores (Figure 4b). Most of the least active FDs have no or few functional groups. They have values of BScores of about under 5000.

The pristine fullerenes C_60_ (FD168) BSc. = 4240, C_70_ (FD50) BSc. = 4482, and C_80_H_2_ (FD169) BSc. = 4646 have the lowest binding affinity.

The following are FD93 BSc. = 5036, FD61 BSc. = 5000 and FD160 BSc. = 4486, with –OH groups belonging to *TA659* (aliphatic alcohols).

The paper [58] reported that “the presence of a hydroxyl group and a low logP value leads to the least toxic molecules …” and that the presence of a higher number of hydroxyl fragments in the studied organic molecules reduces the toxicity to the fish species.

In the work of [63], it was shown that pristine C_60_ is not relatively toxic. They found that fullerenes did not cause acute or subacute toxicity in a large number of living organisms. A review of the toxicity of fullerenes was published by Aschberger et al. [6] in which the authors also indicated that pristine fullerenes have low toxicity.

We then built regression and CPANN models (see Table 1) to predict the binding affinity described below.

### 2.6. Regression Models for Predicting Binding Affinities Using Descriptors QPpolrz and TD (Regression Model 1) and Optimal Descriptors DCW (Regression Model 2)

The models for predicting binding affinities were built. Regression model 1 was based on QPpolrz and TD descriptors. Regression model 2 was based on optimal DCW descriptors.

The characteristics of regression model 1 with regression equations for each of the output variables and their statistical performance are shown in Appendix A. We have high statistical performance for predicting Average BScores (coefficient of determination R^2^ = 0.934, binding affinity (R^2^ = 0.906)). The binding affinity of nine proteins associated with aquatic toxicity (1E3K, 1GOS, 1GS4, 1H82, 1OG5, 1UOM, 2F9Q, 2J0D, 3ERT) was predicted, with R^2^ ranging from 0.660 to 0.893. In the case of protein 1D6U, the R^2^ appeared to be low and was 0.366.

The statistical performance of regression model 2 is shown in Appendix A. In this case, R^2^ for Average BScore was equal to 0.926, while the binding affinity was R^2^ = 0.912. For the binding scores of the nine proteins associated with aquatic toxicity (1E3K, 1GOS, 1GS4, 1H82, 1OG5, 1UOM, 2F9Q, 2J0D, 3ERT), R^2^ ranged from 0.624 to 0.867. In the case of 1D6U protein, R^2^ appeared to be low and equal to 0.373. This low coefficient of determination in the case of the 1D6U protein may be attributed to the metal content of the protein and its origin from *E. coli*.

In section Appendix A, we created plots of the actual response versus the predicted response for regression model 1 (Appendix A) and regression model 2 (Appendix A). These plots show how well our models fit and predict each observation, with the points forming a linear pattern, indicating that the models fit the data well and predict the response accurately, with the exception of the models relating to protein 1D6U.

The applicability domain of QSAR models is explained in detail in Appendix A. We used Williams plots to assess the applicability domain of regression model 1. The standardized residuals versus leverage plots are shown in Appendix A. We identified FDs outside the limits, including the warning leverage threshold (h*) and outside the area between ±3 standard deviation units (σ) in regression model 1, and these FDs are listed in Appendix A, with their structures shown in Appendix A.

Similarly, Williams plots for regression model 2 are shown in Appendix A. We identified FDs outside the limits, including the warning leverage threshold (h*) and outside the area between ±3 standard deviation units (σ) in regression model 2. We list these FDs in Appendix A, while their structures are shown in Appendix A.

### 2.7. CPANN Models for Predicting Binding Affinities Using the Descriptors QPpolrz and TD (CPANN Model 1) and the Optimal Descriptors DCW (CPANN Model 2)

QSAR models were developed to predict binding affinity based on the CPANN algorithm. The CPANN model based on QPpolrz + TD descriptors was designated CPANN model 1, whereas the CPANN model based on DCW optimal descriptor was designated CPANN model 2. We used normalized data, and 127 FDs were selected for the training set and 42 FDs for the test set. The leave-one-out cross-validation procedure (LOO-CV) was used to assess the quality and goodness of fit of the models [64,65].

The optimal CPANN model 1 based on QPpolrz and TD descriptors was built with an architecture of 14 × 14 neurons and trained for 400 learning epochs. The optimal CPANN model 2 based on the optimal DCW descriptors was selected with the 14 × 14 architecture and trained for 300 learning epochs.

The statistical performance of the models is presented in Appendix A.

We achieved high statistical performance in predicting the Average BScore in CPANN model 1: coefficient of determination for the training set R^2^ = 0.9839 (RMSE = 0.1264), for the test set Q^2^ = 0.9427 (RMSE = 0.1722), the regression coefficient of leave-one-out cross-validation (LOO-CV) Q^2^cv = 0.9781 (RMSE = 0.1475). In the case of CPANN model 2: squared regression coefficient for training set R^2^ = 0.9702 (RMSE = 0.2324), for test set Q^2^ = 0.9050 (RMSE = 0.2995), leave-one-out cross-validation (LOO-CV) regression coefficient Q^2^cv = 0.9692 (RMSEcv = 0.1752).

For the values of binding scores of nine proteins associated with aquatic toxicity (1E3K, 1GOS, 1GS4, 1H82, 1OG5, 1UOM, 2F9Q, 2J0D, 3ERT), high statistical performance was also obtained in both models (CPANN model 1 and CPANN model 2). For example, in predicting binding affinity for the test set related to these nine proteins, we obtained Q^2^ scores from 0.7752 (for 2JOD) to 0.8968 (for 1H82) in CPANN model 1 and Q^2^ scores from 0.5683 (for 3ERT) to 0.8470 (for 1GOS) in CPANN model 2, except for the models predicting binding affinity for protein 1D6U. In this case, the regression coefficient for the test set (Q^2^ test) appeared to be low and was 0.4205 in CPANN model 1 and 0.3754 in CPANN model 2.

CPANN model 1 based on descriptors QPpolrz and TD has only slightly higher performance than CPANN model 2 based on the optimal descriptor DCW (see Appendix A).

Appendix A illustrates the plots showing the predicted target response (binding affinity) versus the actual response for CPANN model 1. The responses included in the plots are the Average BScore (a), Binding affinity (b), and binding scores (c) for the proteins 1D6U, 1E3K, 1GOS, 1GS4, 1H82, 1OG5, 1UOM, 2F9Q, 2J0D, and 3ERT.

Appendix A shows the plots depicting the predicted target response (binding affinity) versus the actual response for CPANN model 2.

The best results were obtained in the case of the CPANN model based on two descriptors QPpolrz and TD CPANN model 1 with the statistical performance described above.

Appendix A, which contains binding affinity data used in modeling with indication training and test set, provides the opportunity to reproduce the results.

## 3. Materials and Methods

### 3.1. Dataset

The primary dataset of 169 FDs comes from a study published by Ahmed et al. [66]. This dataset includes FDs with different functional groups attached to the fullerene core C_60_, as well as the pristine fullerenes C_60_ (FD168), C_70_ (FD50), and C_80_H_2_ (FD169).

CORAL software (http://www.insilico.eu/coral accessed on 11 September 2023) was used to select training and test sets for future modeling: 127 compounds belong to the training set, while 42 FDs were included in the test set.

The RCSB Protein Data Bank (Burley et al., 2019) [67] is the source for the extraction of ten human proteins (1D6U, 1E3K, 1GOS, 1GS4, 1H82, 1OG5, 1UOM, 2F9Q, 2J0D, 3ERT) that have high similarity to the protein sequences of aquatic species and are therefore associated with aquatic toxicity. These proteins associated with aquatic species are listed in Table 2.

The binding affinity data used in modeling for the dataset of 169 FDs with the training and test set specified are presented in Appendix A.

### 3.2. Descriptors Used in the Study

The 169 FDs were subjected to the calculation and application of various descriptors. In the first section, two descriptors were adopted from the Ahmed et al. [66] study: polarizability (QPpolrz) and topological diameter (TD), which is related to the size of the molecules (see Table 3).

Subsequently, optimal descriptors (DCW) or Monte Carlo descriptors based on the Simplified Molecular Input-Line Entry System (SMILES) were used to represent the molecular structure of the compounds. CORAL software (http://www.insilico.eu/coral accessed on 11 September 2023) was used to calculate these descriptors to translate various information into an endpoint prediction. For reference, see articles [68,69]. The modeling of FDs using these descriptors was discussed in the articles by Toropova et al. [48,70].

The third group of descriptors consisted of twenty-five “drug-like” properties used in this study, and they are listed in Table 3. These descriptors are used to identify structural features and physicochemical properties that might be related to specific biological or pharmacological interactions [71,72,73,74,75]. DataWarrior software (https://openmolecules.org/datawarrior/ accessed on 11 September 2023) (Actelion Pharmaceuticals Ltd., Allschwil, Switzerland) [76] was used to calculate these descriptors with pharmaceutically relevant properties related to transport properties of molecules, ability to bind to proteins, toxic properties, etc.

All the above descriptors were used as input variables in our models and are listed in Table 3.

### 3.3. Characterization of the Parameter Referred to in the Study as Binding Affinity and How It Was Calculated

At the beginning of the study, 169 FDs were optimized using density functional theory (DFT) to obtain the most stable geometry (for details, see references Ahmed et al. [55,66]). These optimized structures were used to obtain binding affinity values using the PatchDock [77] and AutoDockVina software (https://vina.scripps.edu/ accessed on 11 September 2023) [78].

The characteristics of binding affinity and the calculation methods are explained in the article [66] and our previous publication [61].

First, the Average Binding Score (Average BScore) (1) (see Table 4) was taken from our previous work [46]. The Average BScore is calculated as the arithmetic mean of all binding scores for 1117 proteins for each of the 169 FDs.

The binding affinity data used in modeling for the dataset of 169FDs with the indicated training and test set are presented in Appendix A.

Second, binding values for the highest-scoring ligand–protein complexes, termed binding affinity (2) (see Table 4), were calculated by a group at North Dakota University in this study. This is an analog of binding affinity. Binding scores were calculated using the PatchDock software (http://bioinfo3d.cs.tau.ac.il/PatchDock/php.php accessed on 11 September 2023) [77]. The binding score is based on geometric shape complementarity [79]. Protein–ligand docking solutions are sorted by this score. All calculations are performed on a high-performance supercomputing cluster (details can be found in the publication [66]).

Third, binding scores for ten human proteins associated with aquatic species—1D6U (1); 1E3K (2); 1GOS (3); 1GS4 (4); 1H82 (5); 1OG5 (6); 1UOM (7); 2F9Q (8); 2J0D (9); 3ERT (10) (see Table 4)—were obtained using the methods described in the article [66].

The above-mentioned binding affinities (binding scores) as a property were used as output variables in our models.

Finally, binding scores for acetylcholinesterase enzyme (AChE) (PDB code: 1VOT) for *Torpedo californica* (*Tc*AChE) were calculated separately for 169 FDs to determine the potential degree of binding affinity or toxicity of the studied FDs, taking into account the known mechanism of inhibition of *Tc*AChE by the potential toxicants (in our case, FDs) studied.

The binding affinity data used in modeling for the dataset of 169FDs with the training and test set specified are presented in Appendix A.

### 3.4. Regression Analysis

The regression analysis explained in our previous paper [61] was applied in this study to determine the relationship between the independent variables (descriptors derived from structure) and the dependent response (binding affinity). This statistical analysis was based on Minitab software (https://en.freedownloadmanager.org/users-choice/Download_Minitab_14_Version_64_Bit.html accessed on 11 September 2023) [80] to find structural features (expressed by descriptors) that have the greatest influence on binding affinity and to determine the correlations between input and output variables.

The applicability domain of QSAR models has been presented in this study and is described in detail in Appendix A. The theoretical description can be found in our previous work [61]. The leverage approach was used to define applicability domains [64].

### 3.5. The Counter-Propagation Artificial Neural Network (CPANN) Algorithm and Self-Organizing Kohonen Maps

The architecture of CPANN is shown in Figure 5.

The description of CPANN as a self-organizing mapping method is included in our previous work [61]. It should be emphasized that similar objects are close to each other in a two-dimensional map. As a result, clusters of chemicals with similar structures correspond to similar properties in a two-dimensional map. Therefore, both the output layer and the input layer have exactly the same arrangement of neurons [81,82,83,84].

The neural network architecture shown in Figure 5 consists of input parameters (x_1_–x_n_), which by their nature are vector components related to n descriptors computed for all FDs. These vectors are transformed into an input Kohonen layer containing a top map with the distribution of FDs in a 2D map and the distribution of descriptor values in so-called weight maps (labeled as weight levels 1, 2, 3…n in Figure 5) for all individual descriptors. The input and output layers are two-dimensional maps superimposed on each other. The output layers refer to the property, in our case the binding affinity. In this study, they represent the average binding score, binding affinity, and binding scores for ten proteins associated with aquatic toxicity.

The TRACEANN toolbox for Matlab [85] was used to build predictive CPANN models. The Kohonen mapping method involves the visualization of Kohonen levels. Thus, the Kohonen mapping technique [86] allowed for the visualization of molecules with similar structures, which in our case have similar properties or binding affinities.

### 3.6. The ToxAlerts Web Server for Aquatic Toxicity SA Selection

In this study, we used a ToxAlerts web server described in the publication by Sushko et al. [56]. The ToxAlerts platform is available online at https://ochem.eu/login/show.do?render-mode=full accessed on 11 September 2023. This platform can be used in the early stages of drug discovery to identify potential adverse effects of chemicals that have been selected as drug candidates. The environmental risk of industrially produced chemicals can also be assessed through the ToxAlerts portal. Structural features known as “toxicophores” or “structural alerts” (SAs) can be used to identify compounds with potential toxicity. Screening chemical compounds against known SAs can be used in QSAR models to interpret predictive results and to provide mechanistic interpretation of the models. SAs contain the additional mechanism of action information associated with alerts.

## 4. Conclusions

The manuscript provides detailed mechanistic insight into how the properties of fullerene derivatives (FDs) with different functional groups lead to the behavior of FDs toward aquatic species, using computational tools. This was established by examining the binding affinity values for ten human proteins with high similarity to protein sequences in aquatic species.

Structural aquatic toxicity alerts (SAs) were shown to complement and confirm the aquatic toxicity of the FDs studied, while the values of binding affinity provided information on the degree of toxicity. The SAs contain additional information on the mechanism of action associated with specific alerts.

The binding affinity of FDs toward the acetylcholinesterase enzyme AChE complemented the mechanistic insight of how FDs affect aquatic species, as AChE is a known target of toxins in fathead minnow and *Daphnia magna*, and the mechanism of toxic action here is due to inhibition of AChE.

A high correlation was found between binding activities in terms of average BScore, binding affinity, and binding scores for ten human proteins relevant to aquatic toxicity and for *Tc*AChE.

The QSAR models for predicting the values of binding affinity of FDs were developed in accordance with five principles approved by the Organization for Economic Cooperation and Development (OECD) to be accepted by regulators. This enables the prediction of the binding affinity of unknown FDs and avoids complex and time-consuming calculations. The mechanistic interpretation of the models was based on the determination of the main factors affecting toxicity or binding affinity. The following descriptors were found to contribute most to the protein–ligand binding:Topological diameter (TD) and Molecular Weight (MW) characterizing the size of FD molecules;Total Surface Area (TSA) affecting interaction with the environment;Polarizability volume in cubic Angstroms (QPpolrz) related to electric charge and affecting the strength of interaction with proteins;Rotatable Bonds that affect pharmacodynamics events, resulting from interactions with biological targets such as receptors, enzymes, ion channels, nucleic acids, etc.

The extrapolation of human toxicology studies to environmental species is ongoing in the field of nanotoxicology to develop new criteria for the evaluation of nanomaterial prior to manufacture. The study of the interaction of nanoparticles with proteins should be considered as part of toxicological assessment, which is in line with the European regulatory policy promoting a new methodology for toxicity assessment (NAM).

The most active FDs that could have the greatest impact on aquatic organisms were identified in order to prioritize FDs for future testing. This will help reduce the need for animal testing and inform product development decisions prior to mass production.

## Figures and Tables

**Figure 1 ijms-24-14160-f001:**
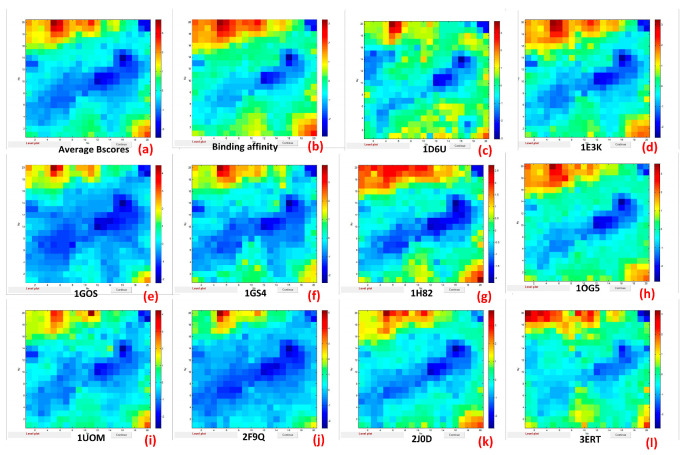
Output maps for Average BScores (**a**), binding affinity (**b**), and binding scores for ten proteins relevant to aquatic species: 1D6U (**c**), 1E3K (**d**), 1GOS (**e**), 1GS4 (**f**), 1H82 (**g**), 1OG5 (**h**), 1UOM (**i**), 2F9Q (**j**), 2J0D (**k**), 3ERT (**l**).

**Figure 2 ijms-24-14160-f002:**
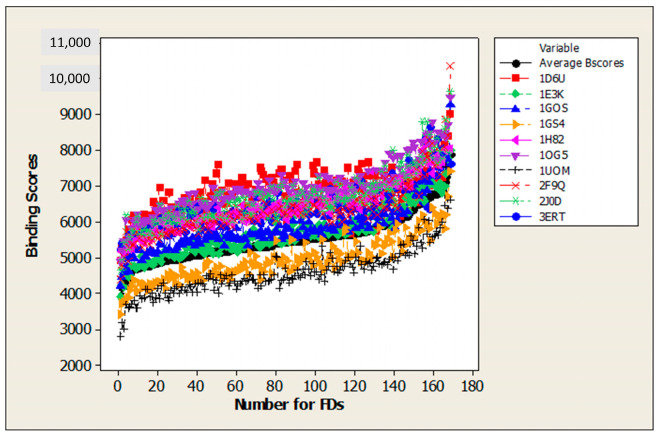
Summary plots comparing binding scores for 10 proteins associated with aquatic toxicity with Average Binding Scores.

**Figure 3 ijms-24-14160-f003:**
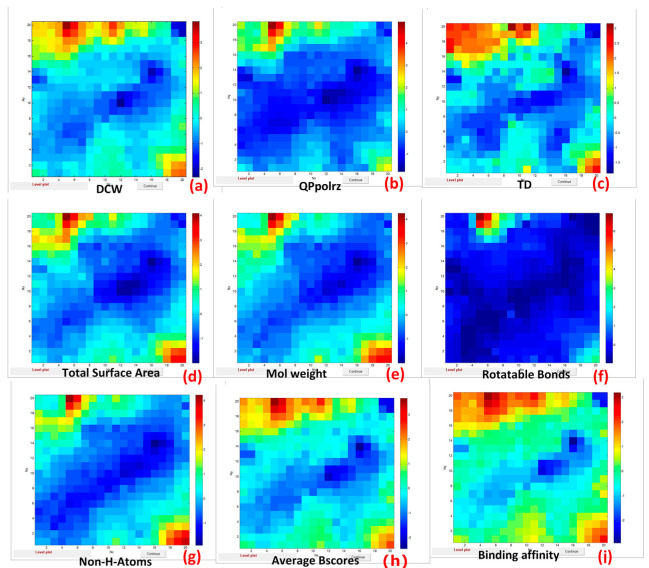
Weight maps for the following descriptors: DCW (**a**), QPpolrz (**b**), TD (**c**), Total Surface Area (**d**), Molecular Weight (**e**), Rotatable Bonds (**f**), Non-H-Atoms (**g**) and output maps for Average BScores (**h**) and binding affinity (**i**).

**Figure 4 ijms-24-14160-f004:**
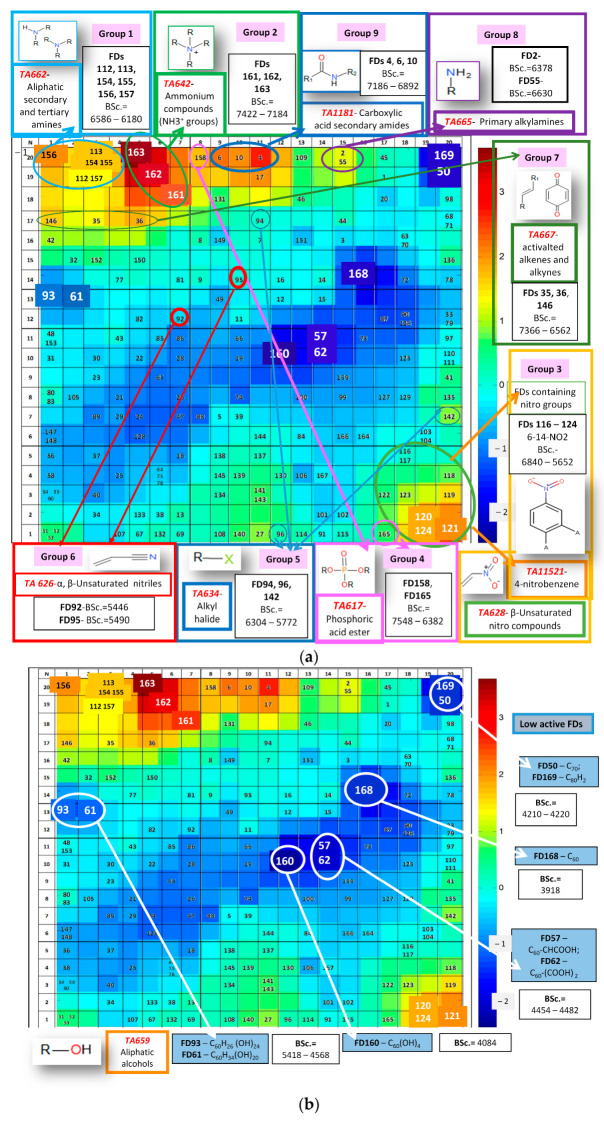
(**a**) Distribution of FDs in the top 20 × 20 map of the CPANN model overlayed, with the output layer showing values of Average BScores with an indication of SAs for aquatic toxicity (TAs) for the most active groups of FDs (1–9). BScores (BSc.) for *Tc*AChE are included here. (**b**) The distribution of FDs in the top 20 × 20 map of the CPANN model overlapping the output layer, with the values of Average BScores for the least active FDs with the indication of structural alerts TAs. BScores (BSc.) for *Tc*AChE are included here.

**Figure 5 ijms-24-14160-f005:**
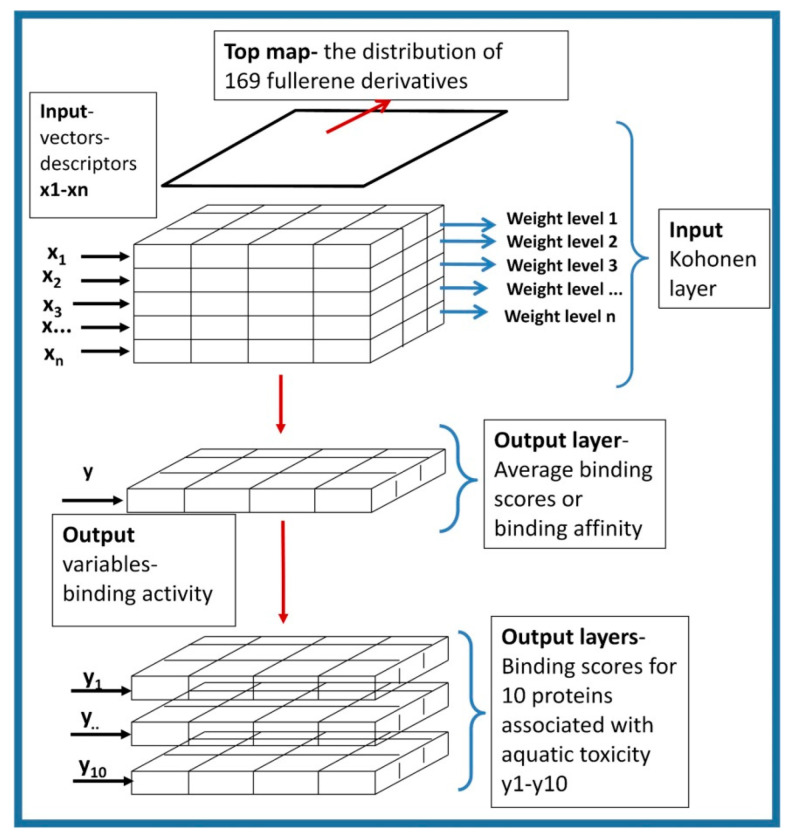
The architecture of CPANN used in this study.

**Table 1 ijms-24-14160-t001:** An overall scheme of the modeling steps, goals and tools applied.

Models	Model Name	Applied Tools	The Goal of the Study
CPANN model based on 28 descriptors (25 drug-like descriptors + QPpolrz + TD + DCW)	ALL_ CPANN model	Visualization in 2D Kohonen map	To find out: (1) Highly correlated variables (2) Significant features affecting binding affinity
Regression model based on 28 descriptors (25 drug-like descriptors + QPpolrz + TD + DCW)	ALL_ regression model	Correlation analysis MLR equations	To find out: (1) Highly correlated variables (2) Significant features affecting binding affinity
CPANN model based on QPpolrz + TD descriptors	CPANN model 1	Prediction QSAR model	Prediction of binding affinity
CPANN model based on DCW optimal descriptors	CPANN model 2	Prediction QSAR model	Prediction of binding affinity
Regression model based on QPpolrz + TD descriptors	Regression model 1	Prediction QSAR model	Prediction of binding affinity
Regression model based on DCW optimal descriptors	Regression model 2	Prediction QSAR model	Prediction of binding affinity

**Table 2 ijms-24-14160-t002:** Ten proteins associated with aquatic species.

PDB_ID	Biochemical Function	Receptor/ (*Protein Code*)	Functional Name of Proteins	Description
1D6U	Neurotransmitter oxidation	Amine oxidase [copper-containing] A (*CuAO*)	Oxidoreductase	Crystal structure of *E. coli* amine oxidase anaerobically red 2 beta-phenylethylamine
1GOS	Neurotransmitter oxidation	Amine oxidase [flavin-containing] B (*MAO-B*)	Oxidoreductase	Human monoamine oxidase B
1H82	Polyamine catabolism	Amine oxidase [flavin-containing] (*PAO*)	Oxidoreductase	Structure of polyamine oxidase in complex with guazatine
1E3K	Endocrine system-related proteins	Androgen receptor (*AR*)	Human progesterone receptor	Human progesterone receptor ligand binding domain in complex 2 with the ligand metribolone (R1881)
1GS4	Endocrine system-related proteins	Androgen receptor (*AR*)	Androgen receptor	Structural basis for the glucocorticoid response in a mutant 2 human androgen receptor (ARccr) derived from an androgen-independent prostate cancer
1UOM	Endocrine system-related proteins	Estrogen receptor (*ESR1*)	Nuclear protein	The structure of estrogen receptor in complex with a selective and potent tetrahydroisochiolin ligand
3ERT	Endocrine system-related proteins	Estrogen receptor (*ESR1*)	Nuclear receptor	Human estrogen receptor alpha ligand-binding domain in complex hydroxytamoxifen
1OG5	Metabolism-related enzyme	Cytochrome P450 2C9 (*CYP2C9*)	Electron transport	Structure of human cytochrome P450 CYP2C9
2F9Q	Metabolism-related enzyme	Cytochrome P450 2D6 (*CYP2D6*)	Oxidoreductase	Crystal structure of human cytochrome P450 CYP2D6
2J0D	Metabolism-related enzyme	Cytochrome P450 3A4 (*CYP3A4*)	Oxidoreductase	Crystal structure of human P450 CYP3A4 in complex with erythromycin

**Table 3 ijms-24-14160-t003:** The list of descriptors used as input variables in the models.

Set1	QPpolrz—Polarizability Volume in Cubic Angstroms (1), TD—Topological Diameter (2);
Set2	DCW-Optimal descriptors (1);
Set3	The group of twenty-five drug-like descriptors: H-acceptors (1), H-donors (2), TSA—total surface area (3), RPSA—relative polar surface area (4), PSA—polar surface area (5), drug-likeness (6), MW—molecular weight (7), cLogP (8), cLogS (9), electronegative atoms (10), stereo centers (11), rotatable bonds (12), rings closures (13), small rings (14), aromatic rings (15), aromatic atoms (16), sp3-atoms (17), symmetric atoms (18), amides (19), amines (20), aromatic nitrogen (21), basic nitrogen (22), acidic oxygen (23), non-H atoms (24), non-C/H atoms (25).

**Table 4 ijms-24-14160-t004:** The list of binding affinities/binding scores used in the study.

Set1	Average Binding Score (Average BScore) (1).
Set2	Binding affinity (2).
Set3	Binding scores for ten proteins associated with aquatic species: 1D6U (1); 1E3K (2); 1GOS (3); 1GS4 (4); 1H82 (5); 1OG5 (6); 1UOM (7); 2F9Q (8); 2J0D (9); 3ERT (10).
Set4	Binding scores for acetylcholinesterase enzyme (AChE) (PDB code: 1VOT) for *Torpedo californica* (*Tc*AChE).

## Data Availability

The new data are available in the Appendix A of this article.

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
