# Peer review of "Cheminformatics and Machine Learning Approaches to Assess Aquatic Toxicity Profiles of Fullerene Derivatives"

_ijms, 2023, doi:10.3390/ijms241814160_

Round 1

Reviewer 1 Report

In this work, authors studied the binding activities of FDs to different proteins using fullerene-specific QSAR applications and machine learning, further predicting their aquatic toxicity. It is significant for ranking FDs in toxicity in vitro, providing references for further FDs applications. However, there are minor issues that must be addressed.

1.     How did the authors choose the kind of fullerene derivatives for computational analysis?

2.     How do active FDs bind to the representative proteins? And why FDs with aromatic nitrogen exhibited much higher binding scores? And what could possibly be causing such a big difference in N-containing and O-containing FDs?

3.     Authors performed computational analysis of 169 FDs and predicted their toxicity, but did these FDs act in vivo? Maybe some simple experimental validation is much better.

4.     What does R in the “Ful R60” mean in figures of SM2?

The Engliash representation should be improved.

Author Response

Dear Editor, Dear Reviewers,

Thank you very much for considering our manuscript.

Answer to Reviewers see in attachment.

Reviewer 2 Report

The main merit of the paper is unambiguously the avoidance of a multitude of experiments on living animals. The authors take an alternative approach by using machine learning methods in order to explore the binding affinities of Fullerene Derivatives (FDs) to human and aquatic organisms proteins and to enzymes, as for example acetylcholinesterase. The data obtained using different theoretical models were utilized to predict the toxicity profiles of 169 FDs. These data can be relevant for the estimation of toxicity of FDs. However, I should like to recommend a validation of at least part of the results by proven methods. The paper is clearly written, and the conclusions are consistent with the data presented. The figures are appropriate in number and show high quality. I should like to recommend at least some experimental spot checks on the obtained results (may be in another paper).

Author Response

(The authors gave the same response as above.)
